# "I feel like I'm in a revolving door, and COVID has made it spin a lot faster": The impact of the COVID-19 pandemic on youth experiencing homelessness in Toronto, Canada

Amanda Noble[1,2]*, Benjamin Owens[2], Naomi Thulien[3,4,5], Amanda Suleiman[2]

1 Factor-Inwentash Faculty of Social Work, University of Toronto, Toronto, Ontario, Canada, 2 Research and Evaluation Department, Covenant House Toronto, Toronto, Ontario, Canada, 3 MAP Centre for Urban Health Solutions, Li Ka Shing Knowledge Institute of St Michael's Hospital, Toronto, Ontario, Canada, 4 Dalla Lana School of Public Health, University of Toronto, Toronto, Ontario, Canada, 5 Centre for Critical Qualitative Health Research, University of Toronto, Toronto, Ontario, Canada

* anoble@covenanthouse.ca

**Data Availability Statement:** Data cannot be shared publicly as participants did not consent to the release of transcripts in a public domain. In

## Abstract

### Purpose

Research has shown that youth experiencing homelessness (YEH) face barriers to social inclusion and are at risk for poor mental health. With the COVID-19 pandemic threatening the health, wellbeing, and economic circumstances of people around the world, this study aims to assess the impacts of the pandemic on YEH in Toronto, Ontario, as well as to identify recommendations for future waves of COVID-19.

### Methods

Semi-structured interviews were conducted with YEH (ages 16–24, n = 45) and staff who work in one of four downtown emergency shelters for youth (n = 31) in Toronto, Ontario.

### Results

YEH experienced both structural changes and psychosocial impacts resulting from the pandemic. Structural changes included a reduction in services, barriers to employment and housing, and changes to routines. Psychosocial outcomes included isolation, worsened mental health, and increased substance use. Impacts were magnified and distinct for sub-populations of youth, including for youth that identified as Black, 2SLGBTQ+, or those new to Canada.

### Conclusions

The COVID-19 pandemic increased distress among YEH while also limiting access to services. There is therefore a need to balance health and safety with continued access to in-

addition, interview data holds sensitive information from youth experiencing homelessness and the staff who serve them in Toronto, Ontario. These populations are small, and thus publicly sharing data would compromise the level of risk of participating in the research—particularly given the marginalization that youth experiencing homelessness already experience and the ways in which staff discussed their places of employment. Researchers who wish to access anonymized interview data are encouraged to contact the University of Toronto's Social Science Research Ethics Board to request access at ethics. review@utoronto.ca, +1 (416) 946-3273.

**Funding:** This work was funded by Making the Shift (MtS) Network of Centres of Excellence, Grant R900206525 (https://makingtheshiftinc.ca). The funding was awarded to AN. The funders had no role in study design, data collection and analysis, decision to publish, or preparation of the manuscript.

**Competing interests:** The authors have declared that no competing interests exist.

person services, and to shift the response to youth homelessness to focus on prevention, housing, and equitable supports for subpopulations of youth.

## Introduction

Research on the impacts of the COVID-19 pandemic has demonstrated the myriad tolls the pandemic has taken on the wellbeing and economic circumstances of people globally [1,2]. These effects have been distributed disproportionately, felt most intensely among those who have been marginalized, including youth experiencing homelessness (YEH). In Toronto, Ontario, Canada, nearly 900 young people between the ages of 16–24 reside in the city's emergency shelter system on any given night [3]. Among YEH in Canada, the disproportionate representation of particular sub-populations is well documented. These sub-populations include youth that identify as two-spirit, lesbian, gay, bisexual, transgender and/or queer (2SLGBTQ+), Indigenous youth, Black youth or youth of colour, newcomer and/or refugee youth [4], and perhaps less-well documented but worthy of further research and discussion, neurodiverse youth or youth with developmental disabilities such as autism spectrum disorder and fetal alcohol spectrum disorder [5,6].

Although there are numerous causes of youth homelessness, the most common are family conflict, abuse, aging out of the child welfare system and, for youth that identify as 2SLGBTQ +, living with non-affirming families [4]. Research has shown that YEH also experience high levels of mental distress, including anxiety, depression, and suicidal ideation, likely the result of trauma and adverse childhood experiences [7]. They face multiple barriers to social inclusion, including maintaining employment and remaining in school, and often have limited social networks and supports [8]. The experience of homelessness, particularly when prolonged, can be traumatic in itself, and can lead to worsened mental health, substance use, and exposure to exploitation and victimization [9,10]. As research has found that the global COVID-19 pandemic has taken a toll on young people's mental health and access to supports, particularly when combined with economic precarity [e.g., 11–14], scholars in the field have drawn attention to the ways these issues could be exacerbated among YEH [15–17].

An incipient body of research since the start of the pandemic has confirmed these hypotheses. Quantitative research found significant rates of poor mental health and increased substance use among this population since the onset of the pandemic [18]. Additionally, a qualitative account uncovered high rates of anxiety and depression related to isolation, uncertainty about the future, and insecurity, among other pandemic-related factors for YEH [19]. Increased substance use was also identified as a coping mechanism [19]. Compounding these issues, both studies reported barriers to accessing supports—including those from government and service providers—as organizations adapted to the new work imperatives of the COVID-19 era [18,19]. Researchers have also reported other losses experienced by young people that interwove with their mental health during the pandemic. Notably, given the economic impacts of restrictions used to curb the spread of COVID-19, unemployment grew over this period, leading to greater financial uncertainty among young people, including those experiencing homelessness [19,20].

Emerging research has found that the impacts of the COVID-19 pandemic are particularly acute among individuals that face additional inequities by virtue of their identities. Results from a large quantitative study in the United States found that mental health issues have disproportionately occurred in "young adults, Hispanic persons, Black persons, essential workers, and those receiving treatment for pre-existing psychiatric conditions" during the pandemic

[21]. Research on transgender and gender-diverse youth during the pandemic found higher rates of adverse mental health compared to young people who are cisgender, as well as greater difficulty accessing services and family supports [14]. Looking at the experiences of 2SLGBTQ + youth more broadly in a mixed-methods study with 61 youth that identified as 2SLGBTQ+, the majority of whom were precariously housed or homeless, the authors reported high rates of poor mental health, presence of non-suicidal self-harm behaviours, increased substance use and suicidal ideation related to the pandemic [22]. An alarming thirty-six percent of study participants reported attempting suicide since the start of the pandemic. Consistent with other research, these youth experienced increased mental health needs alongside decreased access to supports and services.

Our study adds to the existing literature by investigating the experiences of youth living in one of four downtown emergency shelters in Toronto, Ontario during the COVID-19 pandemic. The primary objective was to understand the impact COVID-19 had on YEH, including among youth that moved to a hotel for services and those that remained in shelter, and to identify recommendations for future waves of COVID-19.

## Methods

In this paper we present the qualitative findings of a broader, sequential mixed-method study, where the qualitative findings were used to inform the selection of quantitative tools and generate hypotheses to be tested in the second, quantitative phase.

### Ethics

This study was reviewed and approved by the University of Toronto Research Ethics Board (RIS Protocol No. 39900). In accordance with COVID-19 research protocols, interviews were conducted remotely, and verbal consent was attained subsequent to thoroughly reviewing the informed consent form with participants and prior to the beginning of the interviews. Participants were allowed to withdraw from the research at any time during the interview and were informed that withdrawal would not affect their ability to receive the incentive associated with participation. To maintain anonymity, participants in this study are identified using unique numerical codes, prefixed by 'Y' for youth and 'ST' for staff.

### Participants

Interviews were conducted with 45 YEH, ages 16–24, as well as 31 staff who work in one of four downtown emergency shelters for youth in Toronto, Ontario, Canada. Interviews with youth were conducted between January and March 2021, a period which roughly aligned with the city's third wave of the COVID-19 pandemic. Interviews with staff were conducted slightly earlier, between November and December 2020. Interviews lasted approximately 45 minutes. Recruitment drew on a multifaceted convenience sampling approach, distributing flyers to service providers and disseminating emails to staff who then encouraged participation from both youth and fellow staff members. Snowball sampling was also used, as word of mouth between youth accessing services spurred additional participation. Youth who participated were provided a $50 incentive for their time.

### Data collection

Data was collected through semi-structured interviews, which focused on the various impacts the pandemic had on YEH, including their mental health and wellbeing, education and employment, and experiences accessing services. Staff were asked about their own experiences

providing services during the pandemic, as well as their perspectives on the wellbeing of their clients. Due to health and safety protocols associated with the COVID-19 pandemic, all interviews were conducted remotely using video conferencing technology or by phone. To address accessibility issues for youth who did not have access to stable WIFI or internet-enabled devices, participants were also able to phone into the interview using a local phone number and secure access code generated by the platform. Interviews were sent to a third-party transcription service, who signed a confidentiality agreement, and were transcribed verbatim.

The saturation point was determined using the notion of 'meaning saturation', which occurs when "no further dimensions, nuances, or insights of issues can be found" [23]. Meaning saturation differs from code saturation, or the point in which additional interview data ceases to expand or change the codebook, and instead attends to the depth of understanding of the interview content. Indeed, meaning saturation was assessed as part of the iterative process of data collection and analysis, where coding occurred concurrently with the collection of interview data. Given the heterogeneity of the sample, which included participants who identified with a variety of genders and ethnicities, among other aspects of identity that shape their experiences, a relatively large sample size was needed to achieve saturation [24].

## Data analysis

Interviews were analyzed inductively using thematic analysis [25], with assistance from the qualitative analysis software Quirkos. Analysis was an iterative process which began with a preliminary review of transcripts from the interviews. Preliminary codes were then developed inductively and discussed among the team to ensure consistency and validity of the codes. The research team formulated two codebooks that served as the foundation for further analysis (one for staff interviews and one for youth interviews) and worked collaboratively to code the data, expanding the list of codes in the process. Following this initial coding, the data was revisited to partition codes into sub-codes, deepening researchers' understanding of the data. Data analysis was grounded in Critical Social Theory, which allowed researchers to interrogate the ways participants' intersecting identities (e.g., race, class, gender, and sexual orientation) shaped their experiences, to critique how social and structural hierarchies were implicated in this process, and to consider how the multiple inequities faced by participants could be addressed [26,27].

## Reflexivity

Interviews were conducted by the first and second authors, as well as by two research assistants. All four interviewers had previous experience conducting qualitative interviews, and the second author and research assistants also received additional training from the first author prior to engaging in interviews. The research team was also comprised of researchers from diverse social locations, granting each a different positionality, and therefore degree of 'insider-status' [28]—vis-à-vis participants. Indeed, the research team included representation from different genders, ages, and ethnicities—among other axes of identity—which allowed for a range of perspectives in data analysis. While participants did not have previous relationships with the interviewers, the informed consent process provided an opportunity for interviewers to introduce themselves, identify the research goals, review the research ethics, and build rapport.

## Results

The impacts discussed most frequently by staff and youth have been organized into two large themes: structural changes caused by the pandemic and the psychosocial implications of the

structural changes. The structural changes included changes to the shelter sector, and increased barriers to obtaining employment and housing. These changes resulted in psychosocial effects including increased feelings of isolation and loneliness, mental health challenges, and substance use. Each of these themes is elaborated on below, followed by a discussion of how the pandemic impacted sub-groups of young people, in particular youth that identify as 2SLGBTQ+, Black youth and newcomer youth.

## Sample

Prior to the start of each interview, participants were asked a series of demographic questions. Both the youth and staff samples included participants from a variety of social locations and backgrounds. See Tables 1 and 2 for a for a breakdown of youth and staff sample demographics, respectively.

## Structural changes resulting from the pandemic

**Service and economic impacts.** When the COVID-19 pandemic hit Toronto's emergency homeless-service system, it was readily apparent that congregate shelters were insufficient to

**Table 1. Youth participant demographics.**

| Characteristics | Frequency | Percent |
|---|---|---|
| **Gender** | | |
| Male | 34 | 75.6% |
| Female | 5 | 11.1% |
| Non-binary/genderqueer | 4 | 8.9% |
| Transgender man | 1 | 2.2% |
| Prefer no answer | 1 | 2.2% |
| **Race/ethnicity** | | |
| White | 8 | 17.8% |
| Black | 15 | 33.3% |
| Bi-racial/multi-ethnic | 3 | 6.7% |
| South Asian | 2 | 4.4% |
| West Asian | 4 | 8.9% |
| East Asian | 1 | 2.2% |
| Asian | 2 | 4.4% |
| Latin American | 3 | 6.7% |
| First nations, Métis, Inuit, Indigenous | 4 | 8.9% |
| Prefer not to answer | 3 | 6.7% |
| **Indigenous** | | |
| Yes | 4 | 8.9% |
| No | 40 | 88.9% |
| Prefer not to answer | 1 | 2.2% |
| **2SLGBTQ+** | | |
| Yes | 9 | 20.0% |
| No | 33 | 73.3% |
| Prefer not to answer | 3 | 6.7% |
| **Disability** | | |
| Yes | 15 | 33.3% |
| No | 28 | 62.2% |
| Prefer not to answer | 2 | 4.4% |

**Table 2. Staff participant demographics.**

| Characteristics | Frequency | Percent |
|---|---|---|
| **Age** | | |
| 20s | 4 | 12.9% |
| 30s | 9 | 29.0% |
| 40s | 8 | 25.8% |
| 50s | 8 | 25.8% |
| 60s | 1 | 3.2% |
| Prefer not to answer | 1 | 3.2% |
| **Gender** | | |
| Male | 10 | 32.3% |
| Female | 18 | 58.1% |
| Non-binary/genderqueer | 2 | 6.5% |
| Prefer no answer | 1 | 3.2% |
| **Race/ethnicity** | | |
| White | 18 | 58.1% |
| Black | 11 | 35.5% |
| Southeast Asian | 1 | 3.2% |
| Prefer not to answer | 1 | 3.2% |
| **Indigenous** | | |
| Indigenous | 0 | 0.0% |
| Not Indigenous | 30 | 96.8% |
| Prefer not to answer | 1 | 3.2% |
| **2SLGBTQ+** | | |
| 2SLGBTQ+ | 4 | 12.9% |
| Not 2SLGBTQ+ | 26 | 83.9% |
| Prefer not to answer | 1 | 3.2% |
| **Disability** | | |
| Yes | 4 | 12.9% |
| No | 25 | 80.6% |
| Prefer not to answer | 2 | 6.5% |

ward off a public health catastrophe. Under the guidelines of the provincial government and Toronto Public Health, the City of Toronto's Shelter, Support and Housing Administration Division (SSHA) opened up over 25 temporary spaces, including leasing a series of hotels, in order to move thousands of people to hotels to accommodate social distancing measures [29]. For the downtown youth shelters, this meant moving dozens of residents into hotels so each resident in the hotels and shelters could have their own room.

At the onset of the pandemic, agencies were forced to halt all in-person programming that did not provide an essential service, resulting in numerous service closures and, in other cases, a shift to a virtual format. This led to a dramatic reduction in social services available to YEH, including both internally at their agency and externally in the community. In addition, many of the rules at the shelters were tightened (such as granting nights away from the shelter) particularly during lockdowns, which meant that youth were not able to see friends and family regularly.

In addition to affecting shelter capacity and access to services, COVID-19 restrictions also shaped the economic context for YEH, increasing both employment and housing precarity. As restrictions were imposed over the course of the pandemic, employment opportunities for many young people were severely reduced. Nearly two-thirds of youth participants indicated

that their experiences of employment were negatively affected by the pandemic, including lay-offs, difficulty finding work, and heightened precarity among those who were employed. When asked whether COVID-19 played a role in them coming to the shelter, one youth described how they were laid off at the beginning of the pandemic and were not hired back despite the store reopening months later:

> "I was working full time in a management position. I had a lot of responsibilities for the overall managing of the store, and then I just got laid off out of nowhere, and the store was closed for a few months and then it opened, and they didn't hire me back, and then it closed again." [Y041].

For YEH who were unemployed, finding work was also a challenge. As several youth pointed out, the economic consequences of the pandemic and ensuing restrictions disproportionately affected segments of the economy that typically hire young people, including accommodation, retail, and food service. With these businesses temporarily closed or operating with reduced staffing, employment opportunities were scarce. For one youth, the easing of restrictions—which roughly preceded the pandemic's 'waves'—provided some hope, but the resurgence of the virus made it difficult to secure stable work:

> ". . .It's hard to find anything now, because positions are filled. Not many places are accepting new hires and a lot of services are closed, other than essential. Things opened up for a second wave, and that's when I decided to reapply, but then it went back, so. . ." [Y008].

Among youth who were able to work during the pandemic, the majority were concentrated in precarious jobs, often working in the 'gig economy' as couriers for food delivery apps. One youth who worked in this sector described an oversaturated labour market in which it was hard for him to secure gigs:

> "I have sometimes, just when Uber Eats—like I can't really make as much, because there'll be weeks that I couldn't even make 500 bucks. . . like not even 300, because there is like so many drivers out there." [Y017].

For some youth, the lack of employment opportunities left them feeling "stuck", or unsure how to move forward in their lives as they were unable to save money to obtain housing or leave the shelter. Elaborating on this heightened housing precarity, one youth said:

> "It's definitely been hard because I have no income coming in. It's just difficult, not knowing how I'm going to move forward." [Y002].

In addition to facing difficulties saving for housing, the high levels of unemployment and precarious work among youth in the sample led to some losing their housing alongside work. For one youth, moving to the shelter was a direct result of lost employment after restaurants first shut down:

> "I had my own place; I lived downtown for more than one and a half years, nearly about two years [. . .] 24th March or 25th March [my work] closed due to COVID. . . After that I haven't got any job, so have, I did a program, that's the job training program, but no luck with that too." [Y023].

Prior to coming to shelter, youth and staff also shared how the pandemic encouraged youth to endure unstable and—at times—unsafe living arrangements, often due to a perception of the relative unsafety of the shelter system. As one staff noted, this was particularly prominent among female-identified and trans youth:

"Especially for our . . . for young women and our trans community. They will move in with partners or parents, family systems, because they're so lonely, or they've lost their employment and can't pay their damn rent, that they go back into family systems and relationship systems that are highly toxic, and sometimes physically abusive." [ST012].

For those who sought to move to shelter during the pandemic, some faced obstacles when looking for a shelter space. As previously discussed, shelter capacity was greatly reduced at the start of the pandemic to accommodate social distancing, and other health and safety protocols were put in place that altered how services were delivered. Moreover, in the event of an outbreak (two or more positive COVID-19 cases among residents or staff), agencies were not permitted to accept new intakes until it was declared safe to do so. One youth shared how these changes made it difficult to find a bed: "it was not easy I can tell you that . . . just getting in and trying to find a bed" [Y011].

### Psychosocial implications of structural changes

**Isolation and loneliness.** One of the most frequently discussed impacts of the pandemic by both youth and staff participants was the reduced access to important social networks, and an associated increase in feelings of isolation and loneliness. Due to health and safety protocols intended to stop the spread of COVID-19, many common spaces were closed, in-person group activities were cancelled, and social distancing and single-occupancy bedrooms were enforced throughout the youth shelter system. Many youth also described not being able to see loved ones, including family and friends, and simply staying in their rooms alone all day. For instance, one said:

"I'm a very social person and I love being around people and getting to know people . . . Like right now, because of like everyone's at home, because of lockdown and you can't really like meet people, meet up with friends and stuff, it's . . . it's a very challenging moment, it's testing me, another limit of me." [Y017].

Staff also observed experiences of isolation in the youth they worked with. One staff described experiencing "heartbreak" after hearing one youth say, "'I always knew I was alone, [and] now it really looks like I'm alone'," and described that "This was not the first time I've heard it. . .and it's not the tenth" [ST012], emphasizing how many youth she heard similar sentiments from. Some staff worried that feelings of isolation were particularly prevalent among the youth at the hotel, given the limited number of staff onsite and a layout that was not conducive to socializing. In the words of one staff member, this only worsened under government-mandated lockdown measures: "At the hotel, we're really struggling with young people that are also isolating . . . And with lockdown, that just increases, as well" [ST015]. Connecting youth's tendencies to stay in their rooms with feelings of hopelessness, the same staff member described how a lack of motivation and routine contributed to isolating habits: "They tend to just isolate . . . the big question is, what are they getting up for, right?" [ST015].

**Mental health challenges.** Given the increased unemployment, change to daily routines, and feelings of isolation—on top of continued homelessness—it is not surprising that staff and youth participants consistently mentioned the negative consequences the pandemic has had

on YEH's mental health. These experiences with poor mental health included both worsening mental health issues among those previously diagnosed, as well as a general increase in incidences of anxiety and depression among youth. One staff member described how social and economic aspects of the pandemic also led to decreased mental health because it replicated many of the YEH's histories of trauma:

> "I've had many, many young people tell me that this entire pandemic mimics trauma for them. It mimics how hopeless they felt, how afraid they felt and how alone they felt, and how they had nowhere to go. . .[they say] 'this reminds me exactly of when I was the most alone in my life'. So not having employment opportunities, not having relationships–we know this is a recipe for tremendous stress and a reopening of trauma, which leads to a decline in mental health for sure." [ST012].

Some youth also described seeking hospitalization to cope with their mental health, in part due to a lack of other available options for support. Describing how the effects of isolation intensified mental health issues he experienced, one youth said:

> "When the pandemic started that's when I was–I had to go to the hospital for my mental health . . . I was inside the house, you know, too much that I, I went a little bit crazy." [Y045].

The increase in mental health concerns was described, particularly by staff, as being compounded by a decrease in supports and services during the pandemic. One staff member said:

> "What really impacted our young people was the pulling of resources. Programming. So, they don't have anywhere to go, they don't have anything to do, and they don't really have any connections they can safely seek. And what we started seeing was a direct impact on an increase in anxiety, increase in depression, and increase in [substance] use." [ST011].

In addition to exacerbating pre-existing mental health issues, others described the ways the pandemic led to an increase in anxiety and depression among youth in the shelter system more broadly. For one staff member who observed these mental health impacts throughout the pandemic, these outcomes were situationally dependent, associated with case numbers and restrictions and compounded as the pandemic stretched to the point of interminability: "I have really also seen their wellness go up and down with the numbers and the amounts of lockdowns and restrictions that we've had" [ST012].

Underlying many of the pandemic-related mental health effects was a sense of hopelessness. As the pandemic stretched beyond the first wave with no foreseeable end, youth in the sample —and the staff who worked with them—described how they had a hard time envisioning or working towards their life goals. For one youth, COVID-19 made him feel hopeless about his living situation and prospects of leaving the shelter. When asked where he might move after his time at the shelter, he said:

> "I'm just living day by day at this point, and that's the shitty part. That's what makes me feel like I'm a shelter kid and I hate that. I feel like I'm in a revolving door and COVID definitely made it spin a lot faster." [Y008].

**Increased substance use as a coping mechanism.**   While not discussed by many youth participants, staff described a marked increase in substance use among YEH during the pandemic. For many staff, this increase in substance use was perceived to be related to heightened mental health concerns and boredom:

> "Mental health got really bad in that capacity around isolation, and people got bored and people needed to self-medicate so they would use substances more." [ST003].

Indeed, several staff and three youth noted that people were using substances more, sometimes "all day" simply because there was nothing else to do. One staff suggested that this was even the case for youth who did not use substances prior to the pandemic, as they were "experimenting to see if they were helpful to get through [the pandemic]" [ST017]. In other cases, staff felt that youth who were working to moderate the volume of substances they used prior to the pandemic—due to the loss of employment or other meaningful activities—gave up on this moderation because they "had nothing to stay as clean as possible for" [ST012]. For some youth with more severe addictions, the decrease in daily structure and available supports was described as making it difficult to remain sober. In the words of one staff:

> "[My client is] a severe alcoholic who had a routine to stay sober including going to the gym, going for coffee with their sponsor, going to an agency for support. . .all of that was gone. A lot of things are virtual, but you don't get the human connection. [So they are struggling with] 'how am I not going to drink? I can't go to my AA meeting; I can't go meet my sponsor'." [ST012].

A few staff described an increase in overdoses on site during the pandemic, which is congruent to the opioid crisis facing much of North America, and the toxic drug supply circulating Toronto [30]. One staff member stated, "there have been a lot more deaths from opioids than COVID" [ST018], suggesting that more attention needs to be paid to this crisis, both in the shelter system and beyond. A reduction in access to harm reduction services and supports was observed by staff participants in at least one of the shelters that participated in this study, as the supplies were previously provided in person and were cancelled in light of the pandemic.

## Subpopulations of youth

Staff participants also discussed the additional impacts the pandemic has had on subpopulations of YEH. A few staff, for instance, referred to a gap in the Toronto youth emergency shelter system for Indigenous youth, or the relatively small amount of culturally appropriate or safe spaces for them. One commented that, at their shelter, they do not see a lot of Indigenous youth and felt that this was due to the lack of Indigenous staff employed at the agency. In their words, this staffing gap "doesn't allow for Indigenous youth to have anyone to identify with" [ST019]. One staff who works closely with youth with developmental disabilities commented that the closure of services as well as the shift to virtual services disproportionately affected these youth, as they often cannot be accommodated online. In this research, however, the most frequently discussed sub-populations were Black youth, newcomer youth, and youth that identify as 2SLGBTQ+, and as such are elaborated on below.

**Black youth.**   Multiple staff and a few youth discussed how systemic racism coupled with the pandemic magnified the impact on Black YEH. One staff member said:

"People of Colour already have to deal with structured racism and all the issues that go with that in society. So just imagine, what's happened with COVID has made things worse." [ST020].

As highlighted earlier, the pandemic greatly affected the ability of study participants to secure employment and housing. This was particularly true for Black youth; several staff described how their Black clients face discrimination from landlords, giving examples of youth discussing units by phone or online, only to be told the unit is no longer available when they meet in person and the landlord sees that they are Black. One staff described the experience of a youth facing blatant racism by a landlord:

"I had a client last week who told me that the landlord asked for first and last [rent]. . .when the youth went to get the keys and the landlord realized it was a person of colour because they had done everything online, they said, 'you know, I don't rent to [offensive racist slur]'." [ST021].

In addition, given the changes in daily routines and activities, and associated increase in boredom among youth, it was not uncommon for shelter staff to encourage youth during the warmer months to go outside. This has different implications for Black youth, who are policed differently, both formally and informally, compared to White people [31]. One staff said:

"With nothing to do, even going outside to hang out can cause stress. I have a couple of clients who are young Black men, and they addressed me because staff were telling them 'maybe go for a walk, get some fresh air', and they both came back to me on separate occasions [and said] 'do you understand what it is to hang out if you're a young Black man? You can't just sit somewhere and loiter. That's what's what they call it, it's not chilling, it's loitering'." [ST012].

The pandemic was also the backdrop of a series of public displays of police brutality against unarmed Black citizens, including the murder of George Floyd, and the ensuing protests and public demands for racial justice and police reform. While these events were horrific and attracted widespread condemnation, they were particularly felt by those who identify as Black, as it was described by staff participants as bringing up their own traumatic experiences of racism and experiences with police coupled with the knowledge that they too could experience such tragedy. As one staff elaborated: "From the females it came from a place like that could be my brother. From the males, it was that could be me" [ST022]. This was supported in this study, with Black staff and youth describing the trauma they experienced as a result of these horrific events. One staff member said:

"The psychological trauma, right? You're seeing state sanctioned violence. I mean I know it's in the States, but we know it happens here. We see it, and we feel it." [ST021].

While staff interviewed stated that the events were acknowledged by leadership at their respective organizations, the general sentiment was that it was insufficient and the impact was not (nor could be) understood by White staff and leadership. A few youth commented that the events were not widely discussed in shelter, and one felt that "COVID took precedence while social justice issues were swept under the rug" [Y014].

**Newcomer youth.** The impact of the pandemic was also acutely experienced by newcomer youth in Toronto's shelter system. Most prominently, the process of obtaining status in

Canada was drastically slowed down, and many were left with no access to work or study permits and no way to secure an income. One youth shared his story, including how his impeded access to refugee status was extremely trying on his mental health:

> "This pandemic, it's affecting me a lot. . .because my first time in Toronto, I applied for my refugee claim and the second day, the lockdown started. I have nothing to do. I don't have my working permit, my study permit, nothing. Just read some books and watch some news-. . .In one month it's going to be one year I'm here; I'm not working, I'm not studying. It's affecting me." [Y030].

It is worth noting that even among those who had obtained a work permit or legal status, similar to the experiences described above for Black youth, the intersection of the pandemic with structural racism created increased hardship for racialized newcomers. One youth participant who identified as a Muslim Person of Colour described experiencing racism before he even met with an employer:

> "Although COVID plays a big role in this, me being Muslim, because when they see my name. . .they don't really check my resume. I did this before: I went to the same place, I handed in two resumes, everything the same, just a different name and I got called for the other name–the fake one I gave them." [Y017].

**2SLGBTQ+ youth.** Several staff in this study discussed the effects of the pandemic on youth that identify as 2SLGBTQ+. In particular, staff described increased isolation, citing the reduced access to identity-specific services and safe communities. One staff said:

> ". . .a lot of community services have provided opportunities for that kind of peer support, social groups, recreational groups. All that stuff has been on pause since the pandemic hit. . .So I think they're feeling it the hardest in terms of feeling isolated and disconnected." [ST004].

Expanding on this, a few staff expressed this disconnection was felt in particular among the trans youth they served:

> "Our trans youth. . .there aren't many spaces in our society where they can gather as a mutual people and feel safe to have conversations and be free to express what they need to express. . .All of those kinds of support systems have gone virtual"–some really made great attempts, but isn't the same." [ST012].

Similarly, youth participants who identified as 2SLGBTQ+ described experiences of isolation and expressed a desire for more identity-specific supports. In the words of one YEH who identified as trans, isolation was a particularly challenging aspect of the pandemic: "It's definitely been very isolating unfortunately. Because of course, you can't see people. . . So, it's been hard on my mental health for various reasons" [Y002]. For another youth who identified as non-binary, accessing services over the phone was not always comfortable and, in one instance, forestalled access to gender-affirming support. After describing successfully renewing their health card over the phone, they shared their discomfort at discussing changing their gender on their passport over the phone:

"At the same time, some other services, like information sessions or. . . I wish there was a mix of both in person or online. Because of COVID in-person is not possible. I really wanted to know about some information about changing my sex on my passport to an X, and I just didn't feel comfortable talking about that over the phone, I would prefer to meet in person and talk about it in person." [Y041].

## Discussion

In this paper we showed that the restrictions of the COVID-19 pandemic led to multiple impacts on YEH, resulting in reduced wellbeing among a population already vulnerable to poor mental health as result of their exposure to adverse childhood experiences and the traumatic experience of being homeless [7]. We discussed the structural changes that emerged from the imposed restrictions aimed at stopping the spread of COVID-19, which resulted in increased barriers to finding employment and housing for YEH in this study. Many youth were laid off as a result of the pandemic, and the employment that was available was largely in the 'gig economy', a particularly precarious form of work, where workers lack basic protections, benefits, or wage guarantees given their (mis)classification as independent contractors [32]. This left many feeling that there was no way to move forward in their lives, or no way to leave the shelter or hotel in which they were residing.

In light of structural changes caused by the pandemic, YEH faced profound psychosocial impacts. Youth who had no activities to fill their days were faced with feelings of boredom, which while seemingly innocuous, has been correlated with anxiety, depression, and substance use [33,34]. Yan et al. [35] also found that in the particular context of COVID-19 lockdowns, boredom was related to particularly high levels of emotional distress. In addition, many youth were left without the positive coping strategies they previously utilized, such as exercise and other recreational activities, which have been shown to increase mental wellbeing among YEH, particularly during the pandemic [36]. Both staff and youth participants frequently referenced the increased feelings of isolation and loneliness YEH faced, as well as their declining mental health and increased substance use.

The findings of this study corroborate similar research studies with YEH in Canada and the United States. Indeed, previous quantitative research has found heightened anxiety, depression, substance use, and sleep disturbances among YEH during the pandemic [18,37]. Similarly, a qualitative study with 20 YEH found that youth experienced job loss, isolation, and worsened mental health, particularly depression and anxiety [19]. All three studies also pointed to the reduced access to services and supports for YEH during the pandemic as a contributing factor.

This research also highlighted the additional effects the pandemic had on various subpopulations who disproportionately experience youth homelessness, particularly youth that identify as Black, 2SLGBTQ+, and newcomer youth. Liu & Modir [38] have argued that during times of national chaos and distress, communities of colour often experience secondary trauma, such as the disproportionate impacts of COVID-19. This research also drew attention to the additional trauma and hardship youth (and staff) that identify as Black faced during the pandemic, not only because of their regular encounters with systemic racism, but also as a result of the very public displays of police brutality and murder against Black individuals and the associated Black Lives Matter movement and protests. Research has shown that exposure to traumatic videos such as the police killings of unarmed citizens is linked to both depressive and post-traumatic stress symptoms among adolescents of colour [39], and Black persons in general [40]. It is therefore critical that service providers are aware the additional trauma and

hardship that Black youth have encountered during the pandemic, as well as a result of systemic racism, and are available to provide additional supports to Black youth (and staff).

Staff also expressed concerns about youth that identify as 2SLGBTQ+ as they were faced with reduced access to affirming communities and social supports and thus experienced increased feelings of isolation. Service providers must be particularly vigilant in providing outreach and support to youth that identify as 2SLGBTQ+ as research has shown that 2SLGBTQ + young people are more likely to experience mental health distress, including depression suicidal ideation [41,42] and that these concerns have become particularly acute during the pandemic [22,43]. For instance, Abramovich et al. [22] found that in their sample of 92 2SLGBTQ + youth in the Greater Toronto Area, the vast majority reported experiencing severe anxiety, moderate to severe depression, and increased loneliness as a result of the pandemic.

Finally, the needs of newcomer youth were discussed in this study, mainly how the pandemic prolonged their ability to obtain legal documents and earn an income. Many newcomer youth interviewed discussed how this created profound stress in their lives, as many were trying to raise money to support themselves as well as family members in their country of origin. Even for those with legal status, newcomer youth of colour in this study also spoke about experiencing discrimination when trying to obtain a job, which is also evidenced by other research [44,45]. While few participants spoke about other sub-populations of YEH, notably Indigenous youth and youth with developmental disabilities, their unique needs cannot be overlooked. In fact, this very lack of discussion, given the needs of Indigenous youth and youth with developmental disabilities that are homeless suggests urgent gaps in both research and programing for these young people.

In addition to the structural and psychosocial consequences of COVID-19 on YEH, the closure of services or move to virtual formats meant that, for many, this increase in distress was also met with a decrease in access to supports. While it is vital to respond to the public health crisis, the rise in mental health concerns and acuity, coupled with the inequitable access and varied levels of comfort in receiving virtual services [e.g., 37], also points to the need to ensure balance between containing the virus and ensuring youth have access to some in-person services and supports. It is clear from this research, therefore, that social restrictions, and a shift to online services disproportionately impact the most marginalized, and measures need to be in place to support them when they need it the most.

Finally, it critical to discuss the effects of the pandemic on YEH with a larger view to the socioeconomic context in which they exist [37]. The YEH in this research experienced the outcomes discussed in this paper because of the structural context they are forced to live within, not simply because of individual-level vulnerabilities. The youth in this study lived in poverty, were without permanent housing, and had a limited ability to secure a living wage. These youth faced multiple social exclusions by virtue of their social location, which was compounded for those who are marginalized further by their race, gender identity, sexuality, ability, and citizenship status. The vast majority of the impacts discussed in this research are not new to the experiences of YEH; rather, the pandemic has only served to exacerbate them and create additional barriers. In fact, with the exception of disruption to daily routines, barriers to obtaining housing and employment, feelings of isolation, mental health concerns, and substance use have all been well documented in the literature pre-pandemic [7,46–51]. Perhaps the more pertinent question to ask is not, 'how has the pandemic impacted YEH?' but how has the pandemic further impacted a group who has already experienced a lifetime of multiple and intersecting inequities? More importantly, what will the long-term consequences be on those youth who were caught in the intersecting crises of homelessness and the pandemic?

The pandemic has further highlighted how inadequate housing creates profound health inequities. Indeed, many of the public health directives issued during the pandemic—the stay-

at-home order for instance—are predicated on the assumption that one has a home to stay at. Similarly, maintaining social distance is exceedingly difficult when one must live with dozens of other youth and staff. Scholars from the Canadian Observatory on Homelessness have been leaders in arguing for a vastly different response to youth homelessness in Canada that focuses on prevention, rapid re-housing and long-term sustainable exits from homelessness [46,52,53]. There has been a call for more 'upstream' supports that provide services well before a youth experiences homelessness—including school-based services, family support and services, and housing provided to youth leaving systems such as child welfare and corrections [54–57]. For those whose experience of homelessness cannot be prevented, services grounded in a Housing First philosophy with long-term supports provided in the community as needed are required, which may include mental health services, assistance with education and employment, as well as services geared to increasing a young person's access to family and natural supports (i.e., people not paid to be in a young person's life). The disproportionate impact on vulnerable subpopulations highlights the need for this response to be rooted in equity, diversion, and inclusion, with identity-specific services available as needed. The need for this was clear before the pandemic, as the long-term negative effects of homelessness are well-documented. The pandemic has only served to make a dire situation even more so. It is incumbent upon policy makers, service providers, and researchers to make YEH a priority; not only to respond to their needs during the pandemic, but to prevent and end their experiences of homelessness once and for all.

## Limitations

There are limitations to note about this research. Given the cross-sectional nature of the interviews, which were conducted between November 2020 and March 2021, findings need to be regarded in their temporal context and do not reflect the changing circumstances of future 'waves' of COVID-19, nor the cumulative effects these waves may have had on YEH and shelter staff. As an exploratory qualitative study, the findings are also not generalizable to the broader population and future quantitative and mixed methods research on the impacts of the pandemic on YEH is warranted.

Additionally, restricting participation in the study by conducting interviews exclusively over the phone and using video-conferencing technology—as per COVID-19 safety protocols —may have introduced a degree of selection bias in the sample. Not all YEH have access to phones, reliable Wi-Fi, and/or Wi-Fi-enabled devices, making it more difficult for these youth to participate in the study. Indigenous youth were also underrepresented in the sample of YEH, leading to a dearth of data on their pandemic experiences. Future research, centred on the experiences of Indigenous YEH during the pandemic, is warranted to address this gap in the literature and highlight the experiences of this subpopulation. Similarly, female-identified youth were also underrepresented in the sample, and the ways in which the pandemic may have had differential impacts on female-identified and non-binary YEH are not documented in the present study. Additional research on the gendered impacts of the pandemic for YEH would begin to address this gap.

## Author Contributions

**Conceptualization:** Amanda Noble, Naomi Thulien.

**Data curation:** Amanda Noble, Benjamin Owens.

**Formal analysis:** Amanda Noble, Benjamin Owens.

**Funding acquisition:** Amanda Noble, Naomi Thulien.

**Investigation:** Amanda Noble, Benjamin Owens.

**Methodology:** Amanda Noble, Naomi Thulien.

**Project administration:** Amanda Noble.

**Supervision:** Amanda Noble.

**Validation:** Amanda Noble.

**Writing – original draft:** Amanda Noble, Benjamin Owens.

**Writing – review & editing:** Amanda Noble, Benjamin Owens, Naomi Thulien, Amanda Suleiman.

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
