## [Editor Report · Decision Letter 0]

28 Feb 2022

PONE-D-22-01338“I feel like I’m in a revolving door, and COVID has made it spin a lot faster”: The impact of the COVID-19 pandemic on youth experiencing homelessness in Toronto, CanadaPLOS ONE

Dear Dr. Noble,

Thank you for submitting your manuscript to PLOS ONE. After careful consideration, we feel that it has merit but does not fully meet PLOS ONE’s publication criteria as it currently stands. Therefore, we invite you to submit a revised version of the manuscript that addresses the points raised during the review process.

We look forward to receiving your revised manuscript.

Kind regards,

Muhammad Abdel-Gawad, MD

Academic Editor

PLOS ONE

Journal Requirements:

"This work was funded by Making the Shift (MtS) Network of Centres of Excellence, Grant R900206525."

We note that you have provided funding information. However, funding information should not appear in the Funding section or other areas of your manuscript. We will only publish funding information present in the Funding Statement section of the online submission form. 

"This work was funded by Making the Shift (MtS) Network of Centres of Excellence, Grant R900206525 (https://makingtheshiftinc.ca). The funding was awarded to AN. The funders had no role in study design, data collection and analysis, decision to publish, or preparation of the manuscript."

Additional Editor Comments:

- The manuscript is too long, try to shorten it.

- Use continuous line numbers (do not restart the numbering on each page).

- Table 1 and table 2 are results, not methods. so, remove them from methodology section and add them to results section.
---

## [Author Response · Author response to Decision Letter 0]

4 Apr 2022

Thank you to the editor and reviewers for your thoughtful comments on our manuscript. We address each of the comments in greater detail below. If there are any other comments or points that arise during the review process, please let us know.

Journal Requirements:

a. The manuscript has been revised to conform to PLOS ONE’s style requirements, and the file names have been updated for this resubmission. Apologies for this mistake. 

2. We note that you have provided funding information. However, funding information should not appear in the Funding section or other areas of your manuscript. We will only publish funding information present in the Funding Statement section of the online submission form. Please remove any funding-related text from the manuscript and let us know how you would like to update your Funding Statement. 

a. All funding-related text has been removed from the manuscript. The funding statement provided in the online submission form is accurate and up-to-date, and does not need to be updated.

3. In your Data Availability statement, you have not specified where the minimal data set underlying the results described in your manuscript can be found.

a. We have modified our data availability statement and added the University of Toronto’s Social Science Ethics Review Board as the contact for anonymized data. The revised statement is as follows:

“Data cannot be shared publicly as participants did not consent to the release of transcripts in a public domain. In addition, interview data holds sensitive information from youth experiencing homelessness and the staff who serve them in Toronto, Ontario. These populations are small, and thus publicly sharing data would compromise the level of risk of participating in the research—particularly given the marginalization that youth experiencing homelessness already experience and the ways in which staff discussed their places of employment. Researchers who wish to access anonymized interview data are encouraged to contact the University of Toronto’s Social Science Research Ethics Board to request access at ethics.review@utoronto.ca, +1 (416) 946-3273.”

Additional Editor Comments:

1. The manuscript is too long, try to shorten it. 

a. We have substantially revised the manuscript to reduce the length. As a result, we have reduced the word count of the body text from 8641 words to 6710.

2. Use continuous line numbers (do not restart the numbering on each page).

a. Apologies for this. Continuous line numbers have been added to the manuscript.

3. Table 1 and Table 2 are results, not methods. So, remove them from methodology section and add them to results section.

a. Thank you for pointing out this mistake. Tables 1 and 2 have been moved to the results section.

---

## [Decision Letter · Decision Letter 1]

21 Jul 2022

PONE-D-22-01338R1“I feel like I’m in a revolving door, and COVID has made it spin a lot faster”: The impact of the COVID-19 pandemic on youth experiencing homelessness in Toronto, CanadaPLOS ONE

Dear Dr. Noble,

Thank you for submitting your manuscript to PLOS ONE. After careful consideration, we feel that it has merit but does not fully meet PLOS ONE’s publication criteria as it currently stands. Therefore, we invite you to submit a revised version of the manuscript that addresses the points raised during the review process.

We look forward to receiving your revised manuscript.

Kind regards,

Baltica Cabieses, PhD

Academic Editor

PLOS ONE

Journal Requirements:

Reviewers' comments:

Reviewer's Responses to Questions

**Comments to the Author**

1. If the authors have adequately addressed your comments raised in a previous round of review and you feel that this manuscript is now acceptable for publication, you may indicate that here to bypass the “Comments to the Author” section, enter your conflict of interest statement in the “Confidential to Editor” section, and submit your "Accept" recommendation.

Reviewer #1: (No Response)

Reviewer #2: (No Response)

2. Is the manuscript technically sound, and do the data support the conclusions?

Reviewer #1: Partly

Reviewer #2: Yes

3. Has the statistical analysis been performed appropriately and rigorously? 

Reviewer #1: N/A

Reviewer #2: N/A

4. Have the authors made all data underlying the findings in their manuscript fully available?

Reviewer #1: Yes

Reviewer #2: Yes

5. Is the manuscript presented in an intelligible fashion and written in standard English?

Reviewer #1: Yes

Reviewer #2: Yes

6. Review Comments to the Author

Reviewer #1: Introduction is clear and presents background that justifies the research problem and the relevance of the research. The methodology is clear and rigorously presented, it is suggested to explain an ethical section where the approval of the ethics committee and the verbal informed consent given by the participants are mentioned, given the vulnerability of the study population. The presentation of results and discussion is clear and attractive. When the case of 2SLGBTIQ+ youth is presented, it is recommended to have first-person quotes from their experience, not only focused on the discourse of the shelter workers. It would be interesting to include a reflection from the gender perspective on the women experience who participated less than man in the study. It is suggested to share the limitations of the study in the discussion section if they exist. In the final list of references the year of publication corresponds to the APA style, it is needed to change the references to Vancouver format.

Reviewer #2: Congratulations on a very interesting study that will surely contribute to asses and mitigate the impacts of the COVID-19 pandemic on the youth experiencing homelessness. It is a substantial contribution that the results are detailed according to the differentiated experiencies of subpopulations.

Some minor aspects could be improved in the manuscript, which will be discussed below.

Regarding the sections where the experience of subpopulations is developed, the one that adresses the 2SLGBTQ+, lacks the perspective of the 2SLGBTQ+ population interviewed, as only staff quotes were incorporated.

Also, according to the consolidated criteria for reporting qualitative research, an identification should be added to each quotation, for example, a participant number (protecting the anonimity of the participants).

Along the report, some minor language editing is required, for example, with repeated words like "support" and "impacts".

Finally, a few format aspects should be adressed, like the size of the numbers in the references and the incorporation of page numbers.

7. PLOS authors have the option to publish the peer review history of their article (what does this mean?). If published, this will include your full peer review and any attached files.

Reviewer #1: No

Reviewer #2: No

---

## [Author Response · Author response to Decision Letter 1]

2 Aug 2022

We are very grateful to the editor and reviewers for your thoughtful comments on our manuscript. We address each of the comments in greater detail below and in our 'Response to Reviewers' document. If there are any other comments or points that arise during the review process, please let us know.

Reviewer #1’s comments:

1. It is suggested to explain an ethical section where the approval of the ethics committee and the verbal informed consent given by the participants are mentioned, given the vulnerability of the study population.

a. Thank you for this comment. We have added a subsection on research ethics to the methodology section of the manuscript where we detail the study’s ethics approval and the verbal informed consent process.

2. When the case of 2SLGBTIQ+ youth is presented, it is recommended to have first-person quotes from their experience, not only focused on the discourse of the shelter workers.

a. This is a very important point raised by both reviewers. The section on 2SLGBTQ+ youth’s experiences has been updated to include quotes from 2SLGBTQ+ youth themselves, who describe experiences of isolation and difficulty accessing services.

3. It is suggested to share the limitations of the study in the discussion section if they exist.

a. The manuscript has been updated to include a section on limitations alongside the discussion. This section addresses the study’s limitations and suggests potential areas for future research. 

4. It would be interesting to include a reflection from the gender perspective on the women experience who participated less than man in the study. 

a. Thank you very much for this comment. While an analysis of the gendered impacts of the pandemic for youth experiencing homelessness is important, it is unfortunately a limitation of this study given the interview protocol and sample. We have added this as a limitation in the limitations section of the manuscript and call for future research to address this gap.

5. In the final list of references the year of publication corresponds to the APA style, it is needed to change the references to Vancouver format.

a. The reference list has been updated to be consistent with the Vancouver citation style guide.

Reviewer #2’s comments:

1. Regarding the sections where the experience of subpopulations is developed, the one that addresses the 2SLGBTQ+, lacks the perspective of the 2SLGBTQ+ population interviewed, as only staff quotes were incorporated. 

a. Thank you for this very important comment. As mentioned above, we have added quotations from 2SLGBTQ+ youth to this section, where they share first-hand their experiences of isolation and difficulty accessing supports. 

2. Also, according to the consolidated criteria for reporting qualitative research, an identification should be added to each quotation, for example, a participant number (protecting the anonymity of the participants). 

a. The manuscript has been revised to include unique participant codes for each quotation. Quotes from youth are prefixed by ‘Y’, and quotes from staff are prefixed by ‘ST’. This coding formula is described in the methods section of the manuscript.

3. Along the report, some minor language editing is required, for example, with repeated words like "support" and "impacts". 

a. Thank you for catching this. The manuscript has been revised to minimize repeated words.

4. Finally, a few format aspects should be addressed, like the size of the numbers in the references and the incorporation of page numbers. 

a. The reference list has been edited for consistency in font size.

---

## [Editor Report · Decision Letter 2]

10 Aug 2022

“I feel like I’m in a revolving door, and COVID has made it spin a lot faster”: The impact of the COVID-19 pandemic on youth experiencing homelessness in Toronto, Canada

PONE-D-22-01338R2

Dear Dr. Noble,

We’re pleased to inform you that your manuscript has been judged scientifically suitable for publication and will be formally accepted for publication once it meets all outstanding technical requirements.

Kind regards,

Baltica Cabieses, PhD

Academic Editor

PLOS ONE

---

## [Editor Report · Acceptance letter]

12 Aug 2022

PONE-D-22-01338R2 

“I feel like I’m in a revolving door, and COVID has made it spin a lot faster”: The impact of the COVID-19 pandemic on youth experiencing homelessness in Toronto, Canada 

Dear Dr. Noble:

I'm pleased to inform you that your manuscript has been deemed suitable for publication in PLOS ONE. Congratulations! Your manuscript is now with our production department. 

Kind regards, 

on behalf of

Dr. Baltica Cabieses 

Academic Editor

PLOS ONE